EMBO
Molecular Medicine

# Beneficial microbiome and diet interplay in early-onset colorectal cancer

Zhengyuan Zhou[1,7], Linda Kleis [ID][2,7], Ana Depetris-Chauvin[1,7], Stefanie Jaskulski[3], Victoria Damerell [ID][4], Karin B Michels[3], Biljana Gigic[4], Ute Nöthlings [ID][2✉] & Gianni Panagiotou [ID][1,5,6✉]

## Abstract

**Colorectal cancer (CRC) is the third most commonly diagnosed cancer and the second leading cause of cancer-related deaths worldwide. Although the risk of developing CRC increases with age, approximately 10% of newly diagnosed cases occur in individuals under the age of 50. Significant changes in dietary habits in young adults since industrialization create a favorable microenvironment for colorectal carcinogenesis. We aim here to shed light on the complex interplay between diet and gut microbiome in the pathogenesis and prevention of early-onset CRC (EO-CRC). We provide an overview of dietary risk factors associated with EO-CRC and contrast them with the general trends for CRC. We delve into gut bacteria, fungi, and phages with potential benefits against CRC and discuss the underlying molecular mechanisms. Furthermore, based on recent findings from human studies, we offer insights into how dietary modifications could potentially enhance gut microbiome composition to mitigate CRC risk. All together, we outline the current research landscape in this area and propose directions for future investigations that could pave the way for novel preventive and therapeutic strategies.**

**Keywords** Diet; Early-onset Colorectal Cancer; Gut Microbiome; Prevention; Therapy
**Subject Categories** Cancer; Evolution & Ecology; Microbiology, Virology & Host Pathogen Interaction

## Introduction

Over the past decades, the incidence of late-onset CRC (LO-CRC) (≥50 years) has generally decreased or remained stable, primarily due to secondary prevention strategies such as undergoing a screening colonoscopy, which is recommended in most countries from the age of 50 onwards (Boardman et al, 2020). In contrast, the global incidence and prevalence of early-onset CRC (EO-CRC) (<50 years) have increased at an alarming rate (Eng et al, 2022).

According to the Global Burden of Disease 2019 study, the incidence of early-onset nasopharyngeal cancer (estimated annual percentage change 2.28%), prostate cancer (2.23%), and CRC (1.73%) showed the fastest increasing trends globally from 1990 to 2019. Conversely, early-onset liver cancer recorded the most significant decline (−2.88%) (Zhao et al, 2023). Among digestive system cancers, EO-CRC has surpassed stomach cancer as the most common early-onset cancer, accounting for 36.8% of cases in 2019 (up from 23.3% in 1990) globally, followed by stomach cancer (30.9% in 1990 to 23.5% in 2019), and liver cancer (21.6% in 1990 to 12.8% in 2019) (Zhao et al, 2023). The age-standardized incidence rates of EO-CRC have increased more rapidly in males than in females (Pan et al, 2022).

Moreover, in 2019, EO-CRC was one of the top four early-onset cancers with the highest mortality and disability-adjusted life years (DALYs) globally and had the highest age-standardized death rate in high-middle SDI regions (Zhao et al, 2023). According to the American Cancer Society, CRC has risen from being the fourth-leading cause of cancer death in both men and women under 50 years in the late 1990s to the first in men and second in women (Siegel et al, 2024). EO-CRC appears to be more aggressive, with tumors that are more frequently located in the left colon, are poorly differentiated, exhibit a higher prevalence of signet ring and mucinous histology, and are often diagnosed at more advanced stages (Mauri et al, 2019). Furthermore, the global incidence and mortality rate of EO-CRC is expected to increase in the next decade among Generation Y (born between 1981 and 1994) and Generation Z (born 1995–2009).

The causes of the alarming rise in EO-CRC are largely unknown and likely multifactorial (Fig. 1). Approximately 20% of EO-CRC cases occur in individuals with a genetic predisposition or family history of CRC, while the remaining 80% are typically sporadic (Ahnen et al, 2014; Spaander et al, 2023). Increasing incidence rates in EO-CRC could be attributed to generational differences in lifestyle factors, such as diet and environmental exposures (Stoffel and Murphy, 2020). The global Westernization of diets, sedentary behavior, physical inactivity, and obesity have been identified as key risk factors (Hofseth et al, 2020; Carroll et al, 2022; Puzzono et al, 2021), possibly through inflammatory and metabolic pathways. Worldwide, adult obesity has more than doubled between 1990 and 2022, and adolescent obesity has quadrupled.

[1]Department of Microbiome Dynamics, Leibniz Institute for Natural Product Research and Infection Biology (Leibniz-HKI), Jena, Germany. [2]Institute of Nutritional and Food Sciences-Nutritional Epidemiology, University of Bonn, Friedrich-Hirzebruch-Allee 7, 53115 Bonn, Germany. [3]Institute for Prevention and Cancer Epidemiology, Faculty of Medicine and Medical Center, University of Freiburg, Freiburg, Germany. [4]Department of General, Visceral and Transplantation Surgery, Heidelberg University Hospital, Heidelberg, Germany. [5]Friedrich Schiller University, Faculty of Biological Sciences, Jena, Germany. [6]Friedrich Schiller University, Jena University Hospital, Jena, Germany. [7]These authors contributed equally: Zhengyuan Zhou, Linda Kleis, Ana Depetris-Chauvin. ✉E-mail: noethlings@uni-bonn.de; gianni.panagiotou@leibniz-hki.de

**Glossary**

| | |
|---|---|
| Colorectal cancer | Commonly known as bowel cancer. A malignancy that originates in the colon or rectum, often developing from pre-cancerous polyps. |
| Early-Onset Colorectal Cancer | Colorectal cancer diagnosed in individuals younger than 50. |
| Gut microbiome | A diverse community of microorganisms, including bacteria, fungi, viruses, and other microbes, residing in the human gastrointestinal tract. |
| Commensal bacteria | Non-harmful, naturally occurring bacteria within the human body. |
| Gut dysbiosis | An imbalance in the composition of the gut microbiome, often associated with reduced microbial diversity and linked to various diseases, such as inflammation and metabolic disorders. |
| Bacteriophages or phages | Viruses that infect and replicate within bacteria. |
| Apoptosis | A programmed cell death process crucial for removing damaged or unwanted cells. |
| Tumor microenvironment | The surrounding cellular and molecular environment in which a tumor exists, including immune cells, blood vessels, and signaling molecules. |
| Chronic intestinal inflammation | Prolonged inflammation within the intestines, often resulting from immune dysregulation. |
| Cell cycle arrest | A regulatory mechanism that halts cell division, typically in response to DNA damage or other stress signals, to prevent the propagation of mutations and maintain cellular integrity. |
| Gut barrier dysfunction | It occurs when the intestinal barrier is compromised, leading to increased permeability that allows pathogens and toxins to enter the bloodstream. |
| Immune homeostasis | The balanced state of immune system activity that maintains defense against pathogens while preventing excessive inflammation and autoimmune reactions. |
| Western diet | A type of usual diet that consists of high consumption of fatty, sugary, and refined products, such as soft drinks, processed meat, and bakery products, and a generally low intake of vegetables, fruits, and dietary fibers. |
| Dietary fiber | A group of complex carbohydrates that has different beneficial effects on human health and is naturally a part of foods such as whole grains, legumes, and vegetables. |
| FFQ (food frequency questionnaire) | A dietary assessment tool used to collect information on long-term dietary habits by inquiring about the usual frequency of consumption of food items over the past 12 months, optionally with questions about portion size. |
| Prospective cohort study | Observational study in which a group of people who represent a defined population but differ in a specific characteristic (risk factors, e.g., diet) are followed over time, and the incidence of a specific outcome (e.g., a disease) is compared between groups. |
| Case-control study | Observational study in which people with and without a certain condition (e.g., a disease) are recruited, and potential risk factors that might have caused the condition are assessed and compared. |
| Cross-sectional study | An observational study in which information about a group of people is collected at a single point in time. In this case, risk factors (e.g., obesity) and the outcome (e.g., a disease) are measured simultaneously. |
| Epidemiological study | Studies that attempt to find an association between human health (e.g., a disease) and a specific cause (e.g., diet) by observing a human population. Examples include prospective cohort studies, case-control studies, and cross-sectional studies. |
| Clinical trial | A carefully designed research study to evaluate the effect of new treatments and tests on human health. Clinical trials are exhaustively reviewed and require pre-approval to be conducted. |

Obesity, a chronic complex disease, is increasing faster among children and young people than among adults (World Obesity Atlas 2023), and both obesity and aging are associated with increased inflammation.

Moreover, chronic inflammation can promote tumorigenesis and its progression, making cancer-related systemic inflammation the seventh hallmark of cancer (Hanahan and Weinberg, 2011). Long-standing inflammatory bowel disease (IBD), such as ulcerative colitis, can undergo neoplastic transformation (Low et al, 2019; Ahmad Kendong et al, 2021). Patients with EO-CRC are more likely to have IBD compared to healthy controls, and IBD patients have nearly a threefold higher risk of developing at an earlier age (Gausman et al, 2020).

The total diversity of the microbiome is crucial for human health, and the role of specific microbes has become more apparent over the past decade (Ni et al, 2023; De Vos et al, 2022; Seelbinder et al, 2023; Marfil-Sánchez et al, 2021; Zhang et al, 2021a). Several studies have focused on the role of the gut microbiome in CRC development, the modulation of response to therapy, and post-treatment (Heshiki et al, 2020; Gopalakrishnan et al, 2018; White

and Sears, 2024; Theodoropoulos et al, 2016), but the gut microbiome's role in the ongoing rise of EO-CRC is still not fully understood. Lifestyle changes may promote CRC via alterations in gut microbes. Thus, the gut microbiota is at the intersection of EO-CRC and the aforementioned lifestyle risk factors, acting as a dynamic living sensor of changes in the human body.

We provide here a comprehensive overview of dietary risk factors for EO-CRC, the beneficial interactions between gut microbes and EO-CRC, diet–microbiome interactions, and the challenges in understanding the association between the gut microbiome and diet in EO-CRC.

## Dietary risk factors associated with EO-CRC

The World Cancer Research Fund (WCRF) reported food groups and nutrients associated with CRC (World Cancer Research Fund/ American Institute for Cancer Research. Continuous Update Project Expert Report 2018. Diet, nutrition, physical activity and colorectal cancer. Available at dietandcancerreport.org).

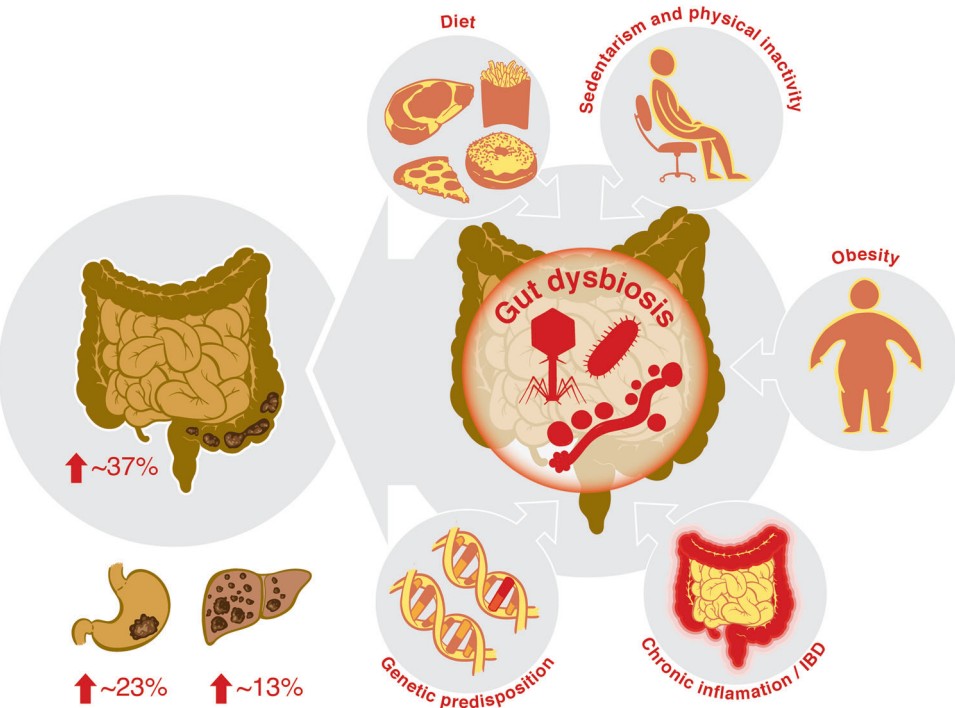

**Figure 1.   Trends in early-onset of digestive cancers and associated factors to CRC.**

Top 3 early-onset digestive cancers and most common risk factors associated with EO-CRC. In the figure, we highlight the role of gut dysbiosis on CRC, including gut bacteria, fungi, and phages. Diet, sedentarism and physical inactivity, obesity, chronic inflammation or IBD, and genetic predisposition modulate the gut microbiome but can potentially also impact CRC development directly.

*Convincing* evidence indicates that processed meat intake is associated with an increased risk of CRC. Furthermore, the intake of whole grains, dairy products, dietary fibers, and calcium supplements is *probably* associated with a decreased CRC risk, while the intake of red meat is *probably* associated with an increased risk of CRC. There is *limited* evidence that suggests an association between the consumption of fish, vitamin D, multi-vitamin supplements, and foods containing vitamin C with a decreased risk of CRC; also, *limited* evidence suggests that a low consumption of fruits and non-starchy vegetables is associated with an increased risk of developing CRC. Notably, age at CRC diagnosis was not explicitly considered in these surveys, rendering inter-pretation difficult as risk factors may, in principle, differ between early-onset and late-onset CRC. Thus, we conducted a systematic literature search in early 2024 in the databases of PubMed and Scopus to identify human studies focusing on the association between diet and EO-CRC risk. Thirteen studies were identified (six prospective cohort and seven case-control studies), whose main characteristics and findings are presented in detail in Appendix Table S1.

Sixteen different dietary factors were identified as potential risk factors for EO-CRC: a Westernized dietary pattern; a higher intake of sugar-sweetened beverages (SSB) or added sugars, red or processed meat, and dairy products; a high-fat diet; and a lower intake of dietary fibers, fish, vegetables, legumes, fruits, beta-carotene, calcium, folate, vitamin C, vitamin D, and vitamin E. Interestingly, some of these dietary risk factors have already been reported for other gastrointestinal diseases, such as IBD, suggesting

a potential synergistic effect of diet and chronic inflammation as risk factors for developing CRC at an early age. For example, a general Westernized dietary pattern seems to be associated with an increased risk of Crohn's disease, while a high sugar and low dietary fiber intake is associated with ulcerative colitis (Levine et al, 2018). In addition, higher dietary fiber, vegetable, and fruit consumption has been reported to be protective against IBD development (Trakman et al, 2022).

A comparison of putative dietary risk factors for EO-CRC with those for CRC for all age groups (WCRF report) showed some similar trends. Higher dietary fiber and calcium intake are associated with a decreased risk of both CRC and EO-CRC, while consumption of red and processed meat increases both the CRC and EO-CRC risk. The *limited* evidence suggesting that the intake of vitamin C, vitamin D, and fish decreases the risk of CRC, and that low consumption of vegetables and fruits increases CRC, is consistent with findings in EO-CRC studies (Table 1). Interestingly, *probable* evidence that the consumption of dairy products decreases the CRC risk contrasts with a report on EO-CRC, although the latter consisted of a small single-center case-control study that only considered the frequency of consumption and didn't control for any confounder (Puzzono et al, 2021). Notably, there is no conclusive evidence for high intakes of added sugar, SSB, fructose, and fried foods, and low intakes of legumes, folate, vitamin E, and beta-carotene as potential influencing factors for CRC due to a low number of available studies and inconsistent findings; nevertheless, some associations of these food groups and nutrients with EO-CRC could be identified. Hence, there may be differences between risk

**Table 1.  Overview of dietary factors associated with EO-CRC risk identified in epidemiological studies.**

| | Study type + Dietary assessment | Findings | References |
|---|---|---|---|
| **Dietary patterns** | | | |
| *Western diet* | Prospective cohort study; validated, semi-quantitative FFQ | Western diet positively associated with risk of EO-adenoma | (Zheng et al, 2021) |
| | Case-control study; FFQ | Western diet associated with increased risk of EO-CRC (Q4 vs. Q1, OR 1.92, CI 1.01–3.66) | (Chang et al, 2021) |
| **Food groups** | | | |
| *Sugar-sweetened foods and drinks* | Prospective cohort study; validated, semi-quantitative FFQ + high school FFQ | Consumption of ≥2 SSB/day in adulthood had doubled the risk of EO-CRC compared to <1 SSB/week (RR 2.18, CI 1.10–4.35) | (Hur et al, 2021) |
| | Prospective cohort study; high school FFQ | High sugar/SSB consumption during adolescence positively associated with risk of adenoma | (Joh et al, 2021) |
| | Case-control study; validated, semi-quantitative FFQ | Highest vs. lowest sweet food intake significantly associated with increased EO-CRC risk (OR 2.70, CI 1.89–3.86) | (Deng et al, 2023) |
| | Case-control study; FFQ | ≥7 vs. <1 SSB drinks per week associated with an increased risk of EO-CRC (OR 2.99, CI 1.57–5.68) | (Chang et al, 2021) |
| *Processed meat and red meat* | Prospective cohort study; validated, semi-quantitative FFQ + high school FFQ | Higher sulfur microbial diet scores associated with increased risk for EO- adenomas (OR Q4 vs.Q1 1.31, CI 1.10–1.56) | (Nguyen et al, 2021) |
| | Case-control study; validated, semi-quantitative questionnaire | Consumption of meat and processed meat has a significant positive association with EO-CRC | (Puzzono et al, 2022) |
| | Case-control study; FFQ | EO-CRC associated with higher red meat consumption (OR 1.10, CI 1.04–1.16) | (Archambault et al, 2021) |
| | Case-control study; validated FFQ | OR of EO-CRC were 1.56 for the highest tertile of processed meat consumption | (Rosato et al, 2013, 201) |
| *Fish* | Case-control study; FFQ interview | EO-CRC associated with lower fish consumption; monthly vs. weekly (HR 1.64, CI 1.01–2.67) | (Pan et al, 2023) |
| | Case-control study; validated FFQ | OR of EO-CRC were 0.78 for the highest tertile of fish consumption | (Rosato et al, 2013) |
| *Fried foods and high-fat diet* | Case-control study; validated, semi-quantitative FFQ | Highest vs. lowest fried food intake significantly associated with increased EO-CRC risk (OR 2.16, CI 1.29–3.62) | (Deng et al, 2023) |
| | Case-control study; questionnaire | Individuals with a high-fat diet had a 98% higher chance for EO-CRC compared to those with a different diet (OR 1.98, CI 1.13–3.49) | (Khan et al, 2015) |
| *Dairy products* | Case-control study; validated, semi-quantitative questionnaire | Consumption of dairy products (5 vs. 3 times/week) has a significant positive association with EO-CRC | (Puzzono et al, 2022) |
| *Fruits* | Case-control study; validated FFQ | OR of EO-CRC were 0.75 for the highest tertile of fruit consumption | (Rosato et al, 2013) |
| *Vegetables and legumes* | Case-control study; validated FFQ | OR of EO-CRC were 0.40 for the highest tertile of vegetables consumption | (Rosato et al, 2013) |
| **Nutrients** | | | |
| *Calcium* | Prospective cohort study; validated, semi-quantitative FFQ | Inverse association between total calcium intake and EO-CRC (HR per 300 mg/day increase 0.87, CI 0.75–1.00) | (Kim et al, 2023) |
| | Case-control study; FFQ | Low calcium intake and EO-CRC risk for colon (OR 1.15, CI 1.05–1.26) | (Archambault et al, 2021) |
| | Case-control study; FFQ | Calcium supplement use associated with reduced risk of EO-CRC (OR 0.53, CI 0.31–0.92) | (Chang et al, 2021) |
| *Vitamin D* | Prospective cohort study; validated, semi-quantitative FFQ | Higher total vitamin D intake significantly associated with reduced EO-CRC risk (HR for ≥450 IU/day vs <300 IU/day 0.49, CI 0.26– 0.93) | (Kim et al, 2021) |
| *Folate* | Case-control study; FFQ | Lower folate intake and EO-CRC risk for colon (OR 1.14, CI 1.04–1.24) | (Archambault et al, 2021) |
| | Case-control study; validated FFQ | OR of EO-CRC were 0.5 for the highest tertile of folate intake | (Rosato et al, 2013) |
| *Dietary fibers* | Case-control study; FFQ | Lower total fiber intake and EO-CRC risk for rectum (OR 1.30, CI 1.14–1.48) | (Archambault et al, 2021) |
| *Beta-carotene* | Case-control study; validated FFQ | OR of EO-CRC were 0.52 for the highest tertile of beta-carotene intake | (Rosato et al, 2013) |

**Table 1.** (continued)

| | Study type + Dietary assessment | Findings | References |
|---|---|---|---|
| *Vitamin C* | Case-control study; validated FFQ | OR of EO-CRC were 0.68 for the highest tertile of vitamin C intake | (Rosato et al, 2013) |
| *Vitamin E* | Case-control study; validated FFQ | OR of EO-CRC were 0.38 for the highest tertile of vitamin E intake | (Rosato et al, 2013) |

factors for early- and late-onset CRC, but given the overall scarcity of available evidence, such comparisons should be interpreted with the utmost caution at this stage.

# The beneficial role of the gut microbiome in EO-CRC

The human microbiota is known to play a significant role in maintaining the homeostasis of the gut ecosystem (Cani, 2018; Zhang et al, 2015). In light of the significant impact of dietary patterns and age-related changes on the composition of the gut microbiome (McCallum and Tropini, 2024), investigating the gut microbiome represents an invaluable opportunity to understand the underlying mechanisms of EO-CRC etiology and progression. Indeed, alterations in the gut microbiome contribute to CRC development (McCallum and Tropini, 2024; White and Sears, 2024), and CRC-associated dysbiosis is closely related to both enrichment of carcinogenic microbes and depletion of beneficial species, including *Lactobacillus*, *Bifidobacterium*, and *Streptococcus* (Kvakova et al, 2022). Notably, most studies examining the role of the gut microbiome in CRC overlook distinctions between early- and late-onset patients, which represents a significant knowledge gap regarding the mechanisms explicitly involved in EO-CRC. Moreover, compared to widely reported CRC-promoting microbes, relatively fewer studies have focused on protective species with the potential for CRC interference (Dougherty and Jobin, 2023). Studies in mice and human cell lines enable direct investigation of the potential beneficial effects of candidate species of bacteria, fungi, and viruses from the human microbiota. However, the translatability of these findings to human health should be approached with caution, as these models may not fully capture the complex interactions within the human body. Below, we discuss gut microbiome species with a putative protective role, their associations in human studies in CRC or EO-CRC-specific cohorts, and their proposed molecular mechanism of action (Table 2).

## Protective microbes in CRC

### Bacterial species
*Lactobacillus* bacteria, one of the dominant microorganisms in the human gut, has shown great potential as a protective genus against CRC progression (Wong, 2023). *Lactobacillus* abundance is decreased in feces and tumor tissue of CRC patients (Dai et al, 2018; Sugimura et al, 2022; Elahi et al, 2023). *Lactobacillus gallinarum* and *Lactobacillus helveticus* were reported to be enriched in non-CRC patients in human fecal and tissue samples, respectively (Dai et al, 2018; Elkholy et al, 2023). Similarly, other bacterial species such as *Blautia producta*, *Clostridium butyricum*, *Streptococcus salivarius*, *Carnobacterium maltaromaticum*,

*Lactococcus lactis*, and *Streptococcus thermophilus* were found to be depleted in fecal samples of CRC patients (Dai et al, 2018; Su et al, 2024; Li et al, 2022c), while *Ruminococcus gnavus* is depleted in CRC tumor samples (Alexander et al, 2023), suggesting potentially distinct microbiome communities in human feces and tissues. In addition, *Lactobacillus rhamnosus* reduces abdominal discomfort and diarrhea in CRC patients, and the combined use of *L. rhamnosus* and *Bifidobacterium lactis*, reduces cell proliferation and improves epithelial barrier function, contributing to CRC prevention (Österlund et al, 2007; Rafter et al, 2007).

A handful of studies have considered age at disease onset, allowing the investigation of beneficial species specifically associated with EO- or LO-CRC. *B. producta*, *S. salivarius*, and *S. thermophilus* were reported with higher abundance in age-matched controls (EO-CTR) compared to EO-CRC (Kong et al, 2023), highlighting the underlying beneficial role of these species for EO-CRC inhibition. Interestingly, *S. salivarius* was additionally found to be enriched in LO-CRC compared to age-matched controls (LO-CTR) (Kong et al, 2023), which implies a different association of this species with the CRC phenotype at different ages. Notably, some species with a proposed beneficial role in the human gut are surprisingly enriched in CRC patients. For example, while *Lactobacillus acidophilus* and *Lactobacillus plantarum* were identified to be enriched in LO-CRC fecal samples compared to LO-CTR (Kong et al, 2023), both species are decreased in patients with polyps (Dadashi et al, 2022), and oral administration of encapsulated live *L. acidophilus* helps restore gut balance through enhancing the diversity of gut microbiota in CRC patients (Gao et al, 2015). *Lactobacillus reuteri* and *Ruminococcus bromii* are enriched in EO-CRC patients compared to EO-CTR (Wu et al, 2023; Kong et al, 2023), even though *L. reuteri* can reduce tumor burden in the human gut microenvironment (Han et al, 2023), and *R. bromii* was recently recommended as a positive microbial signature for the survival of CRC patients (Roelands et al, 2023).

Despite its known health-promoting effects in IBD and diabetes (Zhang et al, 2021b; Rodrigues et al, 2022), the role of the commensal bacteria *Akkermansia muciniphila* on CRC is controversial (Faghfuri and Gholizadeh, 2024; Fan et al, 2021). Even though *A. muciniphila* is less abundant in both severe CRC patients and patients with colitis-associated cancer (Faghfuri and Gholizadeh, 2024; Zhang et al, 2019), its abundance is positively correlated with tumor counts and enrichment in tumor tissue (Baxter et al, 2014; Zhu et al, 2023); furthermore, *A. muciniphila* is enriched in EO-CRC in both human feces and tumor tissue, indicating a potential role in EO-CRC pathogenesis (Barot et al, 2024; Adnan et al, 2024). While studies in cell lines and mouse models have suggested a protective role of *A. municiphila* via regulation of the immune system, others have proposed that *Akkermansia* in the tumor microenvironment could help the tumor evade the body's natural immune response, thereby enhancing immune tolerance and supporting tumor survival (Jiang

**Table 2.  Protective gut microbes in CRC.**

| Microbial species | Mechanism | Associated molecules | Targeted pathways/mechanism | Level of evidence | Human studies | References |
|---|---|---|---|---|---|---|
| BACTERIA | | | | | | |
| *Lactobacillus gallinarum* | Apoptosis and proliferation inhibition | Indole-3-lactic acid | Cell proliferation | Cell lines and mouse models | ↓ CRC (feces) | (Sugimura et al, 2022; Fong et al, 2023; Dai et al, 2018) |
| | Immune system regulation | Indole-3-carboxylic acid | CD8+ T cells function | | | |
| *Lactobacillus helveticus* | Apoptosis and proliferation inhibition | Exopolysaccharides | Cell cycle and Cell proliferation | Cell lines | ↑ normal mucosa (tissue) | (Li et al, 2015; Xiao et al, 2020; Elkholy et al, 2023) |
| *Lactobacillus rhamnosus* | Apoptosis and proliferation inhibition | Bax, caspase-3, and p53 | Tumor size | Mouse models | ↓ CRC discomfort; ↓ cell proliferation, improve epithelial barrier function | (Gamallat et al, 2016; Rafter et al, 2007; Österlund et al, 2007) |
| *Lactococcus lactis* | Apoptosis and proliferation inhibition | Nisin | Cell proliferation | Cell lines | ↓ CRC (feces) | (Ahmadi et al, 2017; Su et al, 2024; Jastrząb et al, 2024) |
| | | Arginine deiminase | c-Myc, p70-S6 kinase phosphorylation and cell cycle | | | |
| *Lactobacillus acidophilus* | Apoptosis and proliferation inhibition | Exopolysaccharides | Apoptosis (via Bcl-2 and Bak) | Cell lines | ↑ LO-CRC (feces); ↓ polyps; alter the intestinal microflora and normalize dysbiosis in CRC | (Kim et al, 2010; El-Deeb et al, 2018; Kong et al, 2023; Gao et al, 2015; Dadashi et al, 2022) |
| | Immune system regulation | Pentasaccharide | Apoptotic cells and CD8+ T cells number | Cell lines and mouse models | | |
| *Lactobacillus plantarum* | Apoptosis and proliferation inhibition | Exopolysaccharides | Reactive O$_2$ species | Cell lines | ↑ LO-CRC (feces); ↓ polyps | (Sun et al, 2021; Zhang et al, 2023a; An et al, 2021; Kong et al, 2023; Dadashi et al, 2022) |
| | Immune system regulation | Indole-3-lactic acid | Cholesterol metabolism & CD8+ T cells function | Cell lines and mouse models | | |
| | Microbial interactions and synergistic therapeutic effects | 5-Fluorouracil | Glycolysis, apoptosis. GABA and cell proliferation | Cell lines | | |
| *Lactobacillus reuteri* | Apoptosis and proliferation inhibition | Reuterin | Redox balance, ribosomal biogenesis, and protein translation | Cell lines and mouse models | ↑ EO-CRC (feces); ↓ CRC tumor burden | (Bell et al, 2022; Kahouli et al, 2016; Han et al, 2023; Kong et al, 2023) |
| | | SCFAs | Fatty acid production | Cell lines | | |
| | | Indole-3-lactic acid | IL-17 pathway | Cell lines and mouse models | | |
| *Lactobacillus fermentum* | Apoptosis and proliferation inhibition | Exopolysaccharides | PI3K/AKT pathway & cell cycle | Cell lines and mouse models | ↑ EO-CRC (feces) | (Li et al, 2022c; Kahouli et al, 2016; Kong et al, 2023) |
| | | SCFAs | SCFA production | Cell lines | | |
| *Clostridium butyricum* | Apoptosis and proliferation inhibition | SCFAs and 2° BAs | SCFA and 2° BA production | Cell lines and mouse models | ↓ CRC (feces) | (Chen et al, 2019; Xu et al, 2023; Liu et al, 2020; Dai et al, 2018) |
| | | MYC (proto-oncogenes) | Cell proliferation/metastasis | | | |
| | Immune system regulation | Cytokines including TNF-α and IL-6 | Inflammation | Mouse models | | |

**Table 2.** (continued)

| Microbial species | Mechanism | Associated molecules | Targeted pathways/ mechanism | Level of evidence | Human studies | References |
|---|---|---|---|---|---|---|
| Streptococcus salivarius | Microbial interactions and synergistic therapeutic effects | Bacteriocins salivaricin | Antimicrobial effect against F. nucleatum | In vitro human colon model | ↑ LO-CRC, ↑ EO-CTR (Chinese Cohort- feces); ↓ CRC (Global Cohort-feces) | (Lawrence et al, 2022; Kong et al, 2023; Dai et al, 2018) |
| Streptococcus thermophilus | Apoptosis and proliferation inhibition | β-Galactosidase | Energy homeostasis and cell proliferation | Cell lines and mouse models | ↓ CRC (feces); ↑ EO-CTRL (feces) | (Li et al, 2021; Dai et al, 2018; Kong et al, 2023) |
| Blautia producta | Immune system regulation | Lyso-glycerophospholipids | CD8⁺ T cells function | Cell lines and mouse models | ↑ EO-CTRL (vs. EO-CRC and vs. LO-CRC) (feces) | (Zhang et al, 2023b; Mao et al, 2024; Li et al, 2022d; Kong et al, 2023) |
| | | Cytokines including TNF-α, IL-6 and IL-1β | Inflammation | Cell lines | | |
| Ruminococcus gnavus | Immune system regulation | Lyso-glycerophospholipids | CD8⁺ T cells function | Cell lines and mouse models | ↓ CRC (tumor) | (Zhang et al, 2023b; Alexander et al, 2023) |
| Carnobacterium maltaromaticum | Microbial interactions and synergistic therapeutic effects | 7-dehydrocholesterol | Metabolic interaction w/F. prausnitzii & activation of vitD receptor | Cell lines and mouse models | ↓ CRC (feces) | (Li et al, 2023b; Dai et al, 2018) |
| Bifidobacterium lactis | Microbial interactions and synergistic therapeutic effects | Resistant starch (synergism) | SCFA production | Mouse models | ↓ Cell proliferation and improves epithelial barrier function | (Le Leu et al, 2010; Rafter et al, 2007) |
| Akkermansia muciniphila | Immune system regulation | Acetyltransferase (Amuc_2172) | HSP70, CD8⁺ T cells function, and gut barrier | Cell lines | ↑ EO-CRC (feces and tumor); ↑ CRC (tissue); ↓ severe CRC | (Jiang et al, 2023; Barot et al, 2024; Adnan et al, 2024; Faghfuri and Gholizadeh, 2024; Fan et al, 2021) |
| Ruminococcus bromii | Microbial interactions and synergistic therapeutic effects | Castalagin (synergism) | Tumor size | Cell lines and mouse models | ↑ EO-CRC (age <= 55); signature for CRC survival | (Messaoudene et al, 2022; Wu et al, 2023; Roelands et al, 2023) |
| Odoribacter splanchnicus | Apoptosis and proliferation inhibition | Malic acid | Cell proliferation | Cell lines and mouse models | ↑ CRC & EO-CRC (feces) | (Oh et al, 2021; Png et al, 2022; Kong et al, 2023) |
| **FUNGI** | | | | | | |
| Saccharomyces cerevisiae | Apoptosis and proliferation inhibition | BAX | Caspase 3 and 7 | Cell lines | ↓ CRC (feces) | (Shamekhi et al, 2020; Wang et al, 2024a; Jadid et al, 2023; Li et al, 2020) |
| | Immune system regulation | Proinflammatory mediators | Inflammation | Mouse models | | |
| | Microbial interactions and synergistic therapeutic effects | Curcumin (synergism) | CRC treatment efficiency | Cell lines | | |
| Pichia kudriavzevii | Apoptosis and proliferation inhibition | Exopolysaccharides | AKT-1, mTOR, and JAK-1 | Cell lines | ↓ CRC (feces) | (Saadat, 2020; Lin et al, 2022) |
| Saccharomyces boulardii | Immune system regulation | TNF-α and IL-6 | Inflammation and tumor load | Cell lines and mouse models | ↓ post-operative complications | (Wang et al, 2019; Kotzampassi et al, 2015) |
| Aspergillus kawachii | \ | \ | \ | \ | ↓ CRC (feces) | (Lin et al, 2022) |

**Table 2.** (continued)

| Microbial species | Mechanism | Associated molecules | Targeted pathways/ mechanism | Level of evidence | Human studies | References |
|---|---|---|---|---|---|---|
| **VIRUSES** | | | | | | |
| *Fusobacterium nucleatum* phages | Pathogen regulation | Significantly reduces *F. nucleatum* biofilm mass | | | ↑ CRC (feces) | (Kabwe et al, 2021; Zheng et al, 2019; Dong et al, 2020; Shen et al, 2021) |
| | Pathogen regulation and immune system activation | Inhibits *F. nucleatum* growth and improves chemotherapy treatment ↓ immune-suppression cells amplification in tumor site and activates the immune system | | | | |
| Enterotoxigenic *Bacteroides fragilis* phages | Pathogen regulation and immune system activation | Reduces host-pathogen counts and cytokine IL-8 levels | | | ↑ CRC (feces) | (Bakuradze et al, 2021; Nakatsu et al, 2018) |
| *Parvimonas micra* phages | | \ | \ | \ | ↑ CRC (feces) | (Shen et al, 2021) |
| *Enterococcus faecalis* phages | Pathogen regulation | Disrupts *E. faecalis* biofilms | | | ↑ CRC (feces) | (Kabwe et al, 2021; Nakatsu et al, 2018) |

et al, 2023; Barot et al, 2024). Similarly, *Lactobacillus fermentum* and *Odoribacter splanchnicus* presented anti-tumor effects in experimental models but were found enriched in EO-CRC patients (Li et al, 2022a; Kong et al, 2023; Oh et al, 2021).

In summary, beneficial bacterial species have a close and complex relationship with CRC pathogenesis, showing potentially distinct distributions in the whole gut microbiome and the tumor microenvironment as well as varied enrichment directions in either disease or healthy conditions. Moreover, the specific enrichment of some protective species in EO-CRC suggests a distinct assembly of protective bacterial communities relevant to EO-CRC and LO-CRC phenotypes. Furthermore, it suggests that the role of these species may vary depending on the specific environmental or physiological context within the host.

### Non-bacterial gut residents

Advances in sequencing and bioinformatics methods have ascertained the presence of non-bacterial components such as archaea, fungi, and viruses in the gut ecosystem with the potential capacity to inhibit CRC (Coker, 2022). Current studies have exhibited the emergence of archaea in the human gut, acting as an efficient signature in predicting CRC (Hoegenauer et al, 2022; Chibani et al, 2021). A negative correlation between the archaea methanogen homoaconitase large subunit protein and CRC was observed, indicating a possible protective effect of archaeal leucine biosynthesis against CRC (Mathlouthi et al, 2023). CRC-enriched archaeal species, including *Haloplanus sp. CBA1113*, and *Natrinema sp. J72*, were found to be antagonistic to probiotic *Clostridium* species (Coker et al, 2020), warning of a potential interference with the benefits of protective bacterial species.

Fungi are one of the major components of the gut microbiota. The fungal composition is not as stable as that of bacteria due to its high variation depending on diet and other environmental factors (Hallen-Adams and Suhr, 2017). Previous studies based on fecal metagenomic sequencing revealed a distinct gut mycobiome in CRC patients, uncovering a higher *Basidiomycota*:*Ascomycota* ratio than in healthy people (Coker et al, 2019). Several yeast groups, including *Saccharomyces*, *Kluyveromyces*, and *Pichia* are recognized as able to improve human health (Shamekhi et al, 2020). Among them, *Pichia kudriavzevii*, *Saccharomyces cerevisiae*, and *Aspergillus kawachii* are reported to be depleted in the feces of CRC patients (Lin et al, 2022; Coker et al, 2020; Li et al, 2020). Moreover, administration of a probiotic cocktail containing *Saccharomyces boulardii* significantly decreased postoperative complications after colorectal surgery and improved the immune system of CRC patients (Kotzampassi et al, 2015).

Viruses are known to be a stable part of the gut microbiome, within which bacteriophages are the most common components with diverse function potentials (Coker, 2022; Tobin et al, 2023). Compared to human-infecting viruses with profound tumor-promoting effects on the gut ecosystem (Park et al, 2023; Ksiaa et al, 2015; Hao et al, 2022), the influence of phages on the CRC progression remains comparatively unexplored. Phage predation can alter the composition of the bacterial community (Qu et al, 2023). A substantial variation in the viral communities of CRC patients in fecal samples has been reported (Gao et al, 2021; Chen et al, 2023). *Inovirus* and *Tunalikevirus* infecting enterotoxigenic *Bacteroides fragilis* are the greatest discriminators identified to be enriched in CRC patients compared to healthy controls (Nakatsu et al, 2018). Interestingly, these two genera were reported to

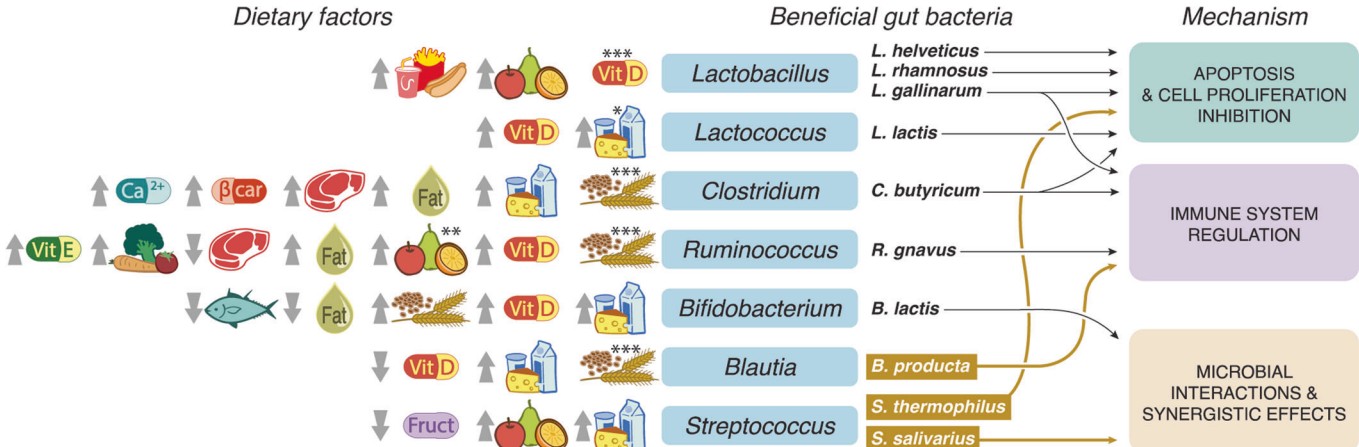

**Figure 2. Dietary factors associated with CRC-protective gut microbes and underlying mechanisms.**

Species highlighted in brown are affected explicitly in EO-CRC patients. The direction of the arrow indicates higher (up) or lower (down) consumption of the specific diet associated with an increased abundance of the indicated bacteria genera. *Higher dairy consumption is associated with an increased abundance of Lactococcus genus but a reduced abundance of species *L. lactis*. **Higher fruit consumption is associated with an increased abundance of Ruminococcus genus but a reduced abundance of species *R. gnavus*. ***Conflicting results are reported in different studies. Fruct: Fructose, and β-car: β-carotene. Disclosure: This figure is the author's interpretation of publications referenced in Tables 2 and 3.

suppress pathogenic *Escherichia coli* (Niu et al, 2014). Moreover, phages targeting *Parvimonas micra*, *Enterococcus faecalis*, and *Fusobacterium nucleatum* are enriched in fecal samples from CRC patients and were proposed as potential biomarkers of CRC (Shen et al, 2021; Nakatsu et al, 2018).

In summary, although substantial evidence supports the protective role of some bacterial species, the role of non-bacterial beneficial species remains comparatively understudied. A meta-analysis of metagenomic samples from 1368 CRC patients and healthy individuals revealed that multi-kingdom biomarkers (bacteria, fungi, and archaea) showed a high performance in predicting the CRC phenotype (Lin et al, 2022). This highlights the importance of exploring microbial inter-kingdom interactions to understand EO-CRC pathology.

## Molecular mechanisms implicated in gut microbiome beneficial effects

In general, the beneficial effects of microbes are primarily mediated through the production of microbial metabolites and other byproducts (Wong, 2023). These substances interact with the tumor tissue and the human immune system, contributing to overall gut health and protection. Studies in mice, cell lines, and human cohorts indicate that these interactions can be implicated in at least three mechanisms with a protective effect against CRC (Fig. 2). In this section, we delve into the mechanisms underlying the beneficial effect of protective bacterial, fungal, and viral species on CRC in general (Table 2) and highlight those mechanisms that could have implications for protection against and/or management of EO-CRC. Only species shown to have a protective role in animal and in vitro models and that are additionally detected in human cohorts are discussed here.

### Apoptosis induction and proliferation inhibition

CRC-protective species have been widely reported to directly induce apoptosis and inhibit the proliferation of CRC tumor cells in

preclinical models in the human gut (Chen et al, 2020; Bell et al, 2022; Li et al, 2022c). *Lactobacillus*, one of the most investigated probiotic bacterial genera, is associated with various molecules triggering apoptosis and inhibiting cell proliferation. Pentasaccharides produced by *L. acidophilus* increase the ratio of the apoptotic cells and provide defense against CRC (El-Deeb et al, 2018). Anti-proliferative exopolysaccharides produced by *L. acidophilus*, *L. fermentum*, and *L. helveticus* induce apoptosis-promoting *Bcl-2* and *Bak* proteins, block the PI3K/AKT signaling pathway, and arrest the tumor cell cycle in G1-phase (Kim et al, 2010; Li et al, 2022a, 2015; Xiao et al, 2020). Furthermore, *L. plantarum*-derived exopolysaccharides increase reactive oxygen species levels and upregulate the expression of the pro-apoptotic proteins such as Bax, caspase-3, caspase-8, and caspase-9, thus contributing to the apoptosis in HT-29 CRC tumor cells (Sun et al, 2021). Apart from exopolysaccharides, *Lactobacillus* can affect CRC tumors through the action of organic acids, antimicrobials, and multiple proteins and enzymes. Short-chain fatty acids produced by *L. fermentum* and *L. reuteri* promote the growth of normal epithelial colon cells and inhibit cell proliferation in the colon cancer cell model Caco-2 (Kahouli et al, 2016; Meenakshi, 2015). *L. gallinarum* suppresses cell proliferation and colony formation via indole-3-lactic acid production, an organic acid with anti-inflammatory properties (Sugimura et al, 2022). *L. reuteri*-derived indole-3-lactic acid exerts anti-tumorigenic effects by downregulating the tumor promotion IL-17 signaling pathway (Han et al, 2023). In addition, *L. reuteri*-produced antimicrobial reuterin induces protein oxidation and translation and inhibits ribosomal biogenesis, decreasing CRC tumor growth and prolonging mice survival (Bell et al, 2022). Likewise, *L. lactis* can prevent CRC progression through its fermentation product nisin, a polycyclic peptide augmenting the apoptotic index at mRNA and protein levels and leading to the death of cancerous cells (Ahmadi et al, 2017). Moreover, *L. lactis* releases arginine deiminase, a cytostatic agent that significantly reduces the growth of colorectal cancer cell models HT-29 and

HCT116 by decreasing the levels of proteins related to cellular growth and contributing to cell cycle arrest (Jastrząb et al, 2024). Oral administration of *L. rhamnosus* in rats reduced tumor incidence and was associated with upregulation of anti-proliferation proteins caspase-7, caspase-9, Bik, Bax, caspase-3, and p53 (Gamallat et al, 2016).

*Clostridium* bacteria are also known to prevent CRC development via SCFA production, which can contribute to decreased proliferation and increased apoptosis in Apcmin/+ mice with a high-fat diet by suppressing the Wnt/β-catenin signaling pathway related to cell proliferation (Chen et al, 2020). In addition, *C. butyricum* was reported to increase SCFAs, decrease fecal secondary bile acids, and activate G-protein coupled receptors, including GPR43 and GPR 109A, promoting tumor cell apoptosis (Chen et al, 2019). Moreover, *C. butyricum* supplementation can destabilize the *MYC* oncogene and suppress CRC cell proliferation by enhancing proteasome-mediated ubiquitination (Xu et al, 2023). *S. thermophilus* can secrete the enzyme β-Galactosidase, which interferes with energy homeostasis and cell cycle arrest in tumor tissue, thereby suppressing CRC cell proliferation (Li et al, 2021). *Odoribacter splanchnicus* showed anti-proliferative activity without inducing cell cycle arrest, and GC-MS analysis suggests that the production of malic acid could mediate this effect (Oh et al, 2021). Further research is needed to confirm a beneficial role for *O. splanchnicus* given its enrichment in CRC patients, particularly EO-CRC (Kong et al, 2023; Png et al, 2022).

In the case of fungal beneficial species, *Pichia kudriavzevii*-produced exopolysaccharides promote apoptosis of CRC cell lines by hindering the AKT-1, mTOR, and JAK-1 pathways (Saadat, 2020). In addition, heat-killed *S. cerevisiae* can induce apoptosis in CRC cell lines by upregulating the expression of Bax and cleaved caspase-3 and caspase-9 proteins (Shamekhi et al, 2020).

### Immune system regulation

The association between chronic intestinal inflammation and CRC has long been established (White and Sears, 2024), and the inflammation status is a significant initiating factor in colorectal tumorigenesis, causing DNA damage, gut barrier dysfunction, and immune system suppression (Wong, 2023). Correspondingly, CRC-protective species can maintain immune homeostasis in the gut ecosystem, thereby alleviating CRC symptoms (Dougherty and Jobin, 2023). A critical effect of these species on immunity is their ability to stimulate the bioactivity of CD8+ T cells, which helps limit CRC progress by killing tumor cells (Raskov et al, 2021). *Lactobacillus* bacteria can produce multiple metabolites that interact with CD8+ T cells. *L. acidophilus* increases the CD8+ T cell percentage and regulates NFκB inflammatory pathway via secretion of pentasaccharides (El-Deeb et al, 2018), while *L. gallinarum* enhances CD8+ T cell function and strengthens anti-tumor immunity through indole-3-carboxylic acid with kynurenine production suppression (Fong et al, 2023). Similarly, *L. plantarum*-produced indole-3-lactic acid ameliorates colorectal tumorigenesis by improving anti-tumor immunity of CD8+ T cells through transcriptionally inhibiting their cholesterol metabolism (Zhang et al, 2023a). On the other hand, *B. producta* and *Ruminococcus gnavus* can degrade lyso-glycerophospholipids in the tissue, inhibiting the activity and maintaining the immune surveillance function of CD8+ T cells and thus protecting the immune function of CD8+ T cells against CRC progression (Zhang et al, 2023b). In addition to metabolites, an outer membrane protein acetyltransferase derived from *A. muciniphila* can improve the intestinal barrier (Wang et al, 2020) and increase the level of heat-shock protein 70 (HSP70), which promotes the anti-tumor protective immunity of CD8+ T cells (Jiang et al, 2023). Despite these promising beneficial effects of *A. muciniphila*, inconsistencies between human cohort studies, along with a reported increased abundance in EO-CRC patients (Barot et al, 2024), warrant a cautious approach to the potential use of *Akkermansia* in the prevention or treatment of EO-CRC.

Another essential effect of beneficial species on the human immune system is their mediation of proinflammatory cytokines. Oral administration of *B. producta* suppresses the increase of inflammatory-stimulated interleukin-6 (IL-6), tumor necrosis factor-α (TNF-α), and interleukin-1β (IL-1β) cytokines and relieves the symptoms of intestinal colitis (Mao et al, 2024). Gavage administration of *C. butyricum* also decreases TNF-α and IL-6 levels, leading to a decreased incidence and size of CRC tumors (Liu et al, 2020). Likewise, *S. boulardii* suppresses fungi-derived cytokines, which reduces tumor load and levels of TNF-α and IL-6 in vivo, preventing ulcerative colitis carcinogenesis (Wang et al, 2019). Moreover, administration of *S. cerevisiae* decreases the expression of multiple cytokines, including IL-1β and IL-6, in murine CRC models (Wang et al, 2024).

### Microbial interactions and synergistic therapeutic effects

Beyond directly participating in anti-tumor activity, beneficial species can indirectly alleviate gut microbiome dysbiosis through community interactions that boost other commensal microorganisms or hinder pathogens with carcinogenic effects. For example, the administration of *Carnobacterium maltaromaticum* reduced intestinal tumor formation in a female-specific manner via the metabolic interaction of *C. maltaromaticum* and *Faecalibacterium prausnitzii*; this metabolic cross-feeding allows the conversion of 7-dehydrocholesterol into vitamin D, which in turn activates mice vitamin D receptors and signals against CRC progression (Li et al, 2023b). In addition, beneficial bacteria can produce chemicals with antibiotic activity and consequently hamper the expansion of pathogenic species in CRC. *S. salivarius* can generate the bacteriocins salivaricin A5 and B, which decrease the number of CRC-pathogenic bacteria *F. nucleatum* in a human distal colon model (Lawrence et al, 2022). Furthermore, synergistic effects with specific diets have also been reported. The berry-derived compound castalagin can physically bind to the cellular envelope of *R. bromii*, and together they inhibit tumor growth, demonstrated in mice (Messaoudene et al, 2022). Moreover, *B. lactis* responds to a resistant-starch diet by increasing SCFA concentrations and contributing to inhibition of CRC tumor progression (Le Leu et al, 2010).

In addition, gut beneficial species can also improve the efficiency of chemopreventive interventions, such as the widely used antimetabolite drug 5-Fluorouracil (5-FU) (Longley et al, 2003), indirectly suppressing CRC tumorigenesis. *L. plantarum*-derived extracellular vesicles and gamma-aminobutyric acid strengthen the effects of 5-FU on anti-proliferation (An and Ha, 2022; An et al, 2021). A similar therapeutics-promoting phenomenon occurs in fungi, where the combination of *S. cerevisiae* and the natural anti-tumor substance curcumin improves the treatment efficiency of CRC in cell lines (Jadid et al, 2023).

In conclusion, multiple molecules, including metabolites, oncogenes, proteins, and antimicrobials, are closely associated with gut species and contribute to the inhibition or prevention of CRC tumor cell growth. Further exploration of the mechanisms involved in non-bacteria beneficial species is needed to better assess their protective capabilities. Notably, most of the mechanisms discussed in this section are derived from cell lines and animal studies that do not distinguish between EO- and LO-CRC, but rather focus on the overall CRC phenotype. Nevertheless, as some bacteria may be involved in CRC pathogenesis both at early and late onset of the disease (Qin et al, 2024), these findings may provide valuable insights for exploring the beneficial role of the gut microbiome in inhibiting EO-CRC. Moreover, the selective enrichment of *S. thermophilus, B. producta,* and *S. salivarius* in EO-CTR compared to EO-CRC (Kong et al, 2023; Dai et al, 2018) highlights a variety of strategies, spanning all three aforementioned anti-tumor mechanisms, that could be explicitly applied to EO-CRC prevention or treatment (Fig. 2). By integrating these strategies, we may develop a multi-faceted approach to manage and treat EO-CRC effectively, leveraging the synergistic effects of a healthy microbiome, targeted probiotics, prebiotics, the immune system, and chemopreventive therapies.

### Beneficial phage infection

Considering the critical role of gut bacteria in CRC tumorigenesis, a growing number of studies report the usage of bacteria-infecting phages as a way of selectively removing pathogens (Wang et al, 2022). *F. nucleatum* phages were found to specifically lyse their hosts and inhibit the growth of the CRC pathogenic bacteria (Zheng et al, 2019). Furthermore, when eliminating *F. nucleatum*, these phages can augment the efficiency of chemotherapy treatments for CRC by enhancing the accumulation of the chemotherapy drugs in tumor tissue, protecting healthy tissues from damage (Zheng et al, 2019). *F. nucleatum* phages could also activate the anti-tumor immunity response for CRC suppression (Dong et al, 2020). Other studies suggest that phages infecting enterotoxigenic *Bacteroides fragilis* reduce both the biomass of the CRC-driven pathogens and the levels of cytokine interleukin-8 (IL-8) in colonic epithelial cells (Bakuradze et al, 2021), suggesting their potential CRC-inhibiting capacity through the alleviation of the inflammatory status. In addition, *Enterococcus faecalis* phages are capable of disrupting bacterial biofilm and reducing the CRC-promoting effects of *E. faecalis* in tumor cells (Kabwe et al, 2021). These studies indicate the versatile roles of beneficial phages against CRC by either limiting the population of gut pathogens, improving chemotherapy efficiency, or regulating the immune system, highlighting the promising future of phage therapy in treating CRC.

# Specific associations of diet and beneficial microbes in CRC

Several studies have reported associations between specific dietary habits and gut bacterial species in CRC. Here, we focus on studies showing associations between dietary risk factors for EO-CRC (Table 1) and their association with the beneficial microbes described above (Table 2). Most are interventional or cross-sectional studies, but some prospective cohort and case-control studies were also identified. The study population consisted of adults in most cases, but the age range differed as some studies focused on a specific group, e.g., elderly people (>65 years) (Ma et al, 2021). In addition, there are several studies on infants (Lei et al, 2021), children (Smith-Brown et al, 2016), or adolescents (Jones et al, 2019). Furthermore, some studies only included either females or males, although most identified studies included both sexes. Participants were primarily recruited from Western countries (e.g., Australia, Europe, or North America) and Asian countries (e.g., China or South Korea), whereas studies involving participants from African or South American countries were scarce. Therefore, broad generalizations should be made cautiously.

Associations between dietary factors identified as able to potentially increase or decrease the risk of developing EO-CRC (Table 1) and gut bacteria species with a proposed beneficial role (Table 2) have been reported (Fig. 2 and Table 3). A diet low in fruits and vegetables has been associated with an increased risk of developing EO-CRC (Rosato et al, 2013), and both higher fruit and vegetable consumption has been associated with an increase in the abundance of the beneficial species *Ruminococcus* (Koponen et al, 2021; Rostgaard-Hansen et al, 2024; Godny et al, 2019). In addition, higher fruit consumption is associated with increased *Streptococcus* abundance (Koponen et al, 2021; Sugimoto et al, 2020). Fruits and vegetables are sources of dietary fibers, which are substrates for fermentation by SCFA-producing bacteria (Wong et al, 2006; Godny et al, 2019), such as *Ruminococcus* spp. and *Streptococcus* spp. Hence, the abundance of those bacteria increases with substrate availability, and the produced SCFAs have multiple beneficial effects on human health, for example on promoting colon epithelium integrity, glucose homeostasis, and the immune system (Koh et al, 2016). Interestingly, while higher fruit consumption has been associated in multiple studies with an increase of *Ruminococcus* genera, Smith-Brown et al find a reduced abundance of *Ruminococcus gnavus* (Smith-Brown et al, 2016), a gut bacteria species with a positive impact on immune system regulation.

Higher consumption of sugar-sweetened foods and drinks (SSBs) and red and processed meat has been associated with an increased EO-CRC risk (Hur et al, 2021; Joh et al, 2021; Deng et al, 2023; Nguyen et al, 2021; Puzzono et al, 2021; and Table 1). While no association between SSB consumption and bacteria species with a potential benefit against EO-CRC has been shown yet, the higher intake of fructose itself appears to be associated with a reduced abundance of *Streptococcus*, including *S. thermophiles* (Jones et al, 2019). Interestingly, most strains of *S. thermophiles* are incapable of fermenting fructose, or they do it very slowly; instead, this species preferably ferments the disaccharides lactose and sucrose (Hutkins and Morris, 1987); hence, large quantities of the monosaccharide fructose could reduce the *S. thermophiles* growth. Moreover, red and processed meat intake has been associated with a decreased abundance of *Ruminococcus* (Farsi et al, 2023). Interestingly, although *Clostridium butyricum* has a putative protective role in CRC, the abundance of *Clostridium sp.* is positively associated with a higher consumption of red meat (Foerster et al, 2014).

Higher dairy product consumption is positively associated with *Clostridium, Streptococcus,* and *Bifidobacterium* (Shuai et al, 2021; Yu et al, 2021), genera that all have a putative protective role against CRC development, and while the association with *Lactococcus* abundance is overall positive, it is negative for *L. lactis* (Swarte et al, 2020). Furthermore, higher dairy consumption is associated with higher levels of *Streptococcus thermophiles* and

**Table 3. Associations between dietary risk factors and beneficial bacteria for CRC.**

| | Dietary exposure | Microbial abundances (genus + species) | References |
|---|---|---|---|
| **Dietary patterns** | | | |
| *Western diet* | Western diet vs. vegan diet | *Lactobacillus* ↑ | (Seel et al, 2023) |
| **Food groups** | | | |
| *Sugar-sweetened foods and drinks* | High vs. low dietary fructose intake | *Streptococcus* ↓ | (Jones et al, 2019) |
| *Processed meat and red meat* | High vs. low (baseline) red meat consumption | *Clostridium sp.* ↑ | (Foerster et al, 2014) |
| | Red + processed meat vs. mycoprotein consumption | *Ruminococcus* ↓ | (Farsi et al, 2023) |
| *Fish* | High vs. low fish consumption | *Bifidobacterium* ↓ | (Viteri-Echeverría et al, 2023) |
| *Fried foods and high-fat diet* | Ketogenic diet (high fat) vs. control | *Clostridium* ↑<br>*Lactococcus* ↑ | (Nakamura et al, 2022)<br>(Nakamura et al, 2022) |
| | High-fat diet (with weight loss) vs. standard diet (lower fat content) | *Ruminococcus 1* ↑ | (Jaagura et al, 2021) |
| | High-fat and low-carbohydrate diet vs. omnivore, vegan or vegetarian diets | *Bifidobacterium* ↓ | (Šik Novak et al, 2023) |
| *Dairy products* | Intake vs. no (low) intake of unpasteurized milk and dairy products | *Lactobacillus* ↑ | (Butler et al, 2020) |
| | High vs. low dairy consumption | *Clostridium* ↑<br>*Streptococcus* ↑<br>*Bifidobacterium* ↑<br>*Bifidobacterium* ↑<br>*Lactococcus* ↑<br>*Streptococcus thermophiles* ↑<br>*Streptococcus salivarius subsp. thermophilus* ↑<br>*Lactococcus lactis* ↓ | (Shuai et al, 2021)<br>(Shuai et al, 2021)<br>(Yu et al, 2021)<br>(Swarte et al, 2020)<br>(Swarte et al, 2020)<br>(Swarte et al, 2020)<br>(Smith-Brown et al, 2016)<br>(Swarte et al, 2020) |
| | High vs. low milk consumption | *Clostridium* ↑<br>*Blautia* ↑<br>*Streptococcus* ↑<br>*Bifidobacterium* ↑<br>*Bifidobacterium* ↑<br>*Bifidobacterium* ↑ | (Shuai et al, 2021)<br>(Li et al, 2018)<br>(Shuai et al, 2021)<br>(Shuai et al, 2021)<br>(Aslam et al, 2021)<br>(Li et al, 2018) |
| | (High) vs. low (no) yogurt consumption | *Streptococcus* ↑<br>*Streptococcus salivarius subsp. thermophilus* ↑ | (Aslam et al, 2021)<br>(Smith-Brown et al, 2016) |
| *Fruits* | High vs. low fruit consumption | *Lactobacillus* ↑<br>*Ruminococcus 1* ↑<br>*Ruminococcus* ↑<br>*Ruminococcus* ↑<br>*Ruminococcus* ↓<br>*Streptococcus* ↑<br>*Streptococcus* ↑<br>*Ruminococcus gnavus* ↓ | (Koponen et al, 2021)<br>(Rostgaard-Hansen et al, 2024)<br>(Koponen et al, 2021)<br>(Godny et al, 2019)<br>(Baldeon et al, 2023)<br>(Koponen et al, 2021)<br>(Sugimoto et al, 2020)<br>(Smith-Brown et al, 2016) |
| | Freeze-dried strawberry consumption vs. usual diet (baseline) | *Bifidobacterium* ↑ | (Ezzat-Zadeh et al, 2021) |
| | Orange juice consumption vs. usual diet (baseline) | *Blautia* ↓ | (Coutinho et al, 2024) |
| *Vegetables* | High vs. low consumption of brassica vegetables | *Clostridium* ↓ | (Kellingray et al, 2017) |
| | High vs. low vegetable consumption | *Ruminococcus 1* ↑ | (Rostgaard-Hansen et al, 2024) |
| **Nutrients** | | | |
| *Calcium* | Calcium supplementation vs. none (placebo) (both with phosphorous) | *Clostridium XVIII* ↑ | (Trautvetter et al, 2018) |
| *Vitamin D* | Vitamin D supplementation vs. placebo | *Lactobacillus* ↑<br>*Ruminococcus YE78* ↑<br>*Blautia* ↓<br>*Bifidobacterium* ↑<br>*Lactococcus* ↑ | (Lei et al, 2021)<br>(Bellerba et al, 2022)<br>(Naderpoor et al, 2019)<br>(Lei et al, 2021)<br>(Kanhere et al, 2018) |
| | Vitamin D supplementation vs. baseline | *Lactobacillus* ↓<br>*Lactobacillus* ↑<br>*Ruminococcus* ↓<br>*Bifidobacterium* ↑<br>*Bifidobacterium* ↑ | (Tabatabaeizadeh et al, 2020)<br>(Schäffler et al, 2018)<br>(Singh et al, 2020)<br>(Singh et al, 2020)<br>(Tabatabaeizadeh et al, 2020) |

**Table 3.** (continued)

| | Dietary exposure | Microbial abundances (genus + species) | References |
|---|---|---|---|
| *Dietary fiber* | Chicory or agave inulin intake vs. placebo | *Bifidobacterium* ↑<br>*Bifidobacterium* ↑<br>*Ruminococcus* ↓ | (Reimer et al, 2020)<br>(Holscher et al, 2015)<br>(Holscher et al, 2015) |
| | Wheat bran consumption vs. usual diet | *Ruminococcus* ↑<br>*Bifidobacterium* ↑<br>*Clostridium XIVa* ↓<br>*Ruminococcus* ↓ | (Aoe et al, 2018)<br>(Granado-Serrano et al, 2022)<br>(Granado-Serrano et al, 2022)<br>(Granado-Serrano et al, 2022) |
| | Rice bran intake vs. placebo | *Lactobacillus* ↑ | (So et al, 2021) |
| | Oat bran intake vs. usual diet | *Bifidobacterium* ↑ | (Xue et al, 2021) |
| | High vs. low dietary fiber intake | *Clostridium* ↑<br>*Ruminoccocus* ↓<br>*Clostridium* ↓<br>*Bifidobacterium* ↑<br>*Ruminococcus* ↓<br>*Blautia* ↑<br>*Clostridium* ↑<br>*Clostridium* ↑<br>*Blautia* ↓<br>*Ruminococcus* ↓<br>*Ruminococcus* ↑<br>*Blautia producta* ↑<br>*Ruminococcus spp* ↓ | (Chen et al, 2013)<br>(Wang et al, 2024b)<br>(Ma et al, 2021)<br>(Hald et al, 2016)<br>(Hald et al, 2016)<br>(Mokhtari et al, 2024)<br>(Mokhtari et al, 2024)<br>(Viteri-Echeverría et al, 2023)<br>(Viteri-Echeverría et al, 2023)<br>(Whisner et al, 2018)<br>(Gomez-Arango et al, 2018)<br>(Um et al, 2023)<br>(Ma et al, 2021) |
| | Soluble dietary fiber intake vs. usual diet | *Bifidobacterium* ↑<br>*Clostridium XIVa* ↓<br>*Ruminococcus* ↓<br>*Ruminococcus* ↓<br>*Clostridium* ↓ | (Granado-Serrano et al, 2022)<br>(Granado-Serrano et al, 2022)<br>(Granado-Serrano et al, 2022)<br>(Holscher et al, 2015)<br>(Zengul et al, 2021) |
| *Beta-carotene* | High vs. low beta-carotene equivalents intake | *Clostridium* ↑ | (Li et al, 2017) |
| *Vitamin E* | High vs. low vitamin E intake | *Ruminococcus* ↓ | (Yan et al, 2023) |

*Streptococcus salivarius* (Swarte et al, 2020; Smith-Brown et al, 2016), two of the three species identified as potentially relevant for EO-CRC prevention (Fig. 2). Notably, *Streptococcus thermophiles* and *Streptococcus salivarious ssp. thermophilus* are often used to start the fermentation in milk during yogurt production, suggesting that their higher presence in the gut could be due to yogurt consumption (Swarte et al, 2020; Smith-Brown et al, 2016). In addition, the consumption of unpasteurized milk is linked to an increased abundance of *Lactobacillus* (Butler et al, 2020), and milk consumption in general is associated with a higher abundance of *Blautia*, *Streptococcus*, *Clostridium*, and *Bifidobacterium* (Li et al, 2018; Shuai et al, 2021). The increase in *Bifidobacterium* abundance could be due to lactose malabsorption, as *Bifidobacterium* species utilize lactose as an energy substrate, which in turn leads to an increase in hydrogen production, potentially explaining the growth of *Blautia* bacteria (Li et al, 2018). Notably, nearly all findings on the association between different dairy products and beneficial gut bacteria emphasize a link between decreased CRC risk and higher dairy consumption.

Higher intake of several nutrients, such as calcium, dietary fibers, or vitamin D, has been associated with a reduced EO-CRC risk (Kim et al, 2023; Archambault et al, 2021; Chang et al, 2021; Kim et al, 2021). The supplementation of calcium or vitamin D seems to impact the gut microbiome positively, increasing the abundance of *Clostridium* with calcium (Trautvetter et al, 2018) and *Bifidobacterium* and *Lactococcus* with vitamin D (Lei et al, 2021; Singh et al, 2020; Tabatabaeizadeh et al, 2020; Kanhere et al, 2018), genera that all have potential benefits against EO-CRC development (Table 2). Mechanisms underlying the impact of vitamin D on the gut microbiome are unclear (Singh et al, 2020),

but calcium can form amorphous calcium phosphate complexes in the human gut, which can impact the gut microbiome and the production of SCFAs (Trautvetter et al, 2018). Moreover, conflicting results on the influence of vitamin D supplementation have been observed for *Lactobacillus* (Lei et al, 2021; Tabatabaeizadeh et al, 2020; Schäffler et al, 2018) as well as for the genus *Ruminococcus* (Bellerba et al, 2022; Singh et al, 2020).

Different sources of dietary fibers, which multiple gut bacteria can ferment to SCFAs through several pathways (Koh et al, 2016), are associated with increased abundances of the genus *Bifidobacterium* (Reimer et al, 2020; Holscher et al, 2015; Granado-Serrano et al, 2022; Xue et al, 2021; Hald et al, 2016). Furthermore, a higher intake of dietary fibers seems to be positively associated with the species of *Blautia producta* (Um et al, 2023), which is beneficial for immune system regulation and is depleted in EO-CRC patients (Table 2). Some studies have shown that a general higher intake of dietary fibers (soluble and insoluble fibers) is positively associated with *Clostridium* abundance (Chen et al, 2013; Mokhtari et al, 2024; Viteri-Echeverría et al, 2023), with soluble fibers being more completely fermented by gut bacteria (Wong et al, 2006); however, while others reported a negative association between *Clostridium* abundance and intake of soluble dietary fibers (Granado-Serrano et al, 2022; Zengul et al, 2021). Similar results have been reported for the genus *Ruminococcus* (Holscher et al, 2015; Aoe et al, 2018; Granado-Serrano et al, 2022; Wang et al, 2024b; Hald et al, 2016; Whisner et al, 2018; Gomez-Arango et al, 2018; Ma et al, 2021). These potentially conflicting results reflect the complex interaction of diet and the abundance of specific gut bacteria species, which may also be influenced by other factors (ethnicity, lifestyle, anthropometric, and more).

Regarding general diets, a Westernized dietary pattern and a high-fat diet have been associated with an increased risk of developing EO-CRC (Zheng et al, 2021; Chang et al, 2021; Khan et al, 2015). Associations between a Westernized diet and CRC incidence were stronger for tumors with elevated levels of pks$^+$ *E. coli* (Arima et al, 2022), a strain that synthesizes the mutagenic compound colibactin (Pleguezuelos-Manzano et al, 2020; Rosendahl Huber et al, 2024). Interestingly, its mutational signature SBS88 can be used to characterize a CRC subtype with better survival (Georgeson et al, 2023). Furthermore, a high-in-fat-but-low-in-carbohydrates diet has been linked with decreased beneficial *Bifidobacterium* abundance (Šik Novak et al, 2023), which supports the described association between a high-fat diet and EO-CRC risk. Notably, the low-carbohydrate component of the diet also entailed a reduced intake of sources of dietary fibers such as grains and starchy vegetables, which in turn could be responsible for the reduced abundance of *Bifidobacterium* species (Jaagura et al, 2021). However, people consuming a high-fat diet showed an increased abundance of *Clostridium*, *Lactococcus*, and *Ruminococcus* (Nakamura et al, 2022; Jaagura et al, 2021), which represents an unexpected association, as members of those genera have been shown to be depleted in CRC patients (Table 2).

The analysis of epidemiological or human intervention studies investigating dietary factors associated with gut microbiome characteristics involved in CRC development opens new avenues, but the findings must be interpreted with careful consideration. Most studies investigating associations between diet and the gut microbiome used 16S rRNA sequencing, which results in findings at the genus level, while studies describing the potential protective role of the gut microbiome have a species-level resolution. This different taxonomic level of identification makes it difficult to compare studies since not all species within a genus may have the same effect on CRC. A comparative analysis of different diet–gut microbiome studies is also complex due to heterogeneity in methodological approaches. For instance, the definition of dietary exposures differed strongly across the intervention studies shown in Table 3. Moreover, while the assessment of dietary exposures in the different prospective cohorts, cross-sectional, and case-control studies were conducted via a variety of FFQs, 24-h recalls, and dietary records and are therefore mostly comparable, differences in quality (validated vs. not validated FFQ) and level of detail (number of items in FFQ; number of days with 24-h recall or dietary record) remain between the studies.

# Current challenges and future directions

Although significant progress has been made, several gaps and challenges persist in the current research landscape. Even with the increasing trend, EO-CRC remains relatively rare compared to CRC in older adults. This makes it challenging to design epidemiological studies on disease incidence. Furthermore, the relative low incidence makes it difficult to recruit sufficient numbers of EO-CRC participants for large-scale prospective studies and clinical trials, potentially limiting the statistical power and generalizability of results. Notably, there are also limitations due to screening age. In particular, the typical screening age of 50 and above will overlook EO-CRC cases, further reducing the pool of eligible participants for studies directed at therapy (Saraiva et al,

2023; Rogers and Johnson, 2021). While some countries have lowered the screening age to 40–45 (Saraiva et al, 2023; Wolf et al, 2018), this change is not yet widespread. Furthermore, since screening guidelines have only recently been updated and traditional screening methods are not typically used for individuals under 45, EO-CRC is often diagnosed at an advanced stage, with approximately 70% of cases being stage III or IV (Rogers and Johnson, 2021).

Several ongoing epidemiological studies are examining the relationship between diet, the microbiome, and EO-CRC. Notable among these are large-scale cohort studies such as the Nurses' Health Study (Everett et al, 2021), the European Prospective Investigation into Cancer and Nutrition (EPIC) (Riboli et al, 2002), and the international ColoCare Study (Ulrich et al, 2019). These studies span diverse populations across the US, Europe, and Asia, including the COLON, CORSA, EnCoRe, DACHS cohorts (Winkels et al, 2014; Gsur et al, 2021; Van Roekel et al, 2014; Chen et al, 2021; Li et al, 2023a, 2022b), and FOCUS consortium (Gigic et al, 2022). In addition, the Multiethnic Cohort (MEC) Study (Fu et al, 2016; Harmon et al, 2017; Tsuzaki et al, 2024), the Shanghai Women's Health Study (SWHS) (Zheng et al, 2005), the Shanghai Men's Health Study (SMHS) (Shu et al, 2015), the Japan Public Health Center-based Prospective Study (JPHC) (Sawada et al, 2020) and the Singapore Chinese Health Study (Hankin et al, 2001) provide valuable data on how diet and lifestyle factors impact EO-CRC risk across different populations. However, these efforts must be accompanied by a commitment to increasing the diversity of study populations. Many of the current studies are conducted predominantly in high-income populations, which may not fully capture the dietary habits, microbiome profiles, and cancer risks of other regions and cultures. Addressing this limitation requires more inclusive research that encompasses diverse ethnic and cultural backgrounds, ensuring that findings are broadly applicable and generalizable on a global scale. Furthermore, despite these advancements, establishing causality requires more than observational data.

Currently, a few large-scale clinical trials are investigating the influence of diet on the gut microbiome in relation to CRC, with none specifically focused on EO-CRC. The VITAL trial (NCT01169259) examines the effects of vitamin D and omega-3 supplements on cancer and cardiovascular disease prevention in over 25,000 participants (Manson et al, 2012). The "Fiber-rich Foods to Treat Obesity and Prevent Colon Cancer" trial (NCT04780477) studies the impact of a high-fiber, legume-rich diet on biomarkers related to CRC risk (Hartman et al, 2024). In addition, the "Modified MAC Diet and Gut Microbiota in CRC Patients" trial (NCT05039060) investigates the effects of a high-fiber diet on the gut microbiome in CRC patients undergoing chemotherapy (Kim et al, 2024). Conducting high-quality prospective studies and clinical trials presents practical and ethical challenges, particularly in ensuring long-term adherence to dietary interventions. Variability in adherence can confound results, necessitating innovative strategies such as regular follow-ups, dietary counseling, and the use of electronic tools for monitoring and feedback.

Looking ahead, one of the most critical advancements in the field will be increasing the resolution with which we study the diet–microbiome–EO-CRC axis. Understanding dietary risk factors throughout childhood, adolescence, and early adulthood and their interaction with the gut microbiome will be important to investigate the complete picture. Currently, most epidemiological

studies are conducted either in adulthood or in children not spanning into adulthood. The DONALD (Dortmund Nutritional and Anthropometric Longitudinally Designed) study (Perrar et al, 2024) exemplifies the critical importance of long-term research focused on young populations. The study underscores the necessity of understanding dietary impacts from a young age, which is particularly relevant for exploring factors contributing to EO-CRC. Furthermore, current research often examines dietary impacts at a broad level, typically focusing on around 150 nutritional components. However, human diets and particularly their associated metabolism involve over 20,000 small molecules, many of which have yet to be thoroughly investigated in the context of CRC and the microbiome. Future micro-level research must delve into these metabolites to better understand their interactions with the gut microbiome. This high-resolution approach will be essential for developing personalized nutrition strategies aimed at preventing EO-CRC. In addition, the consumption of so-called ultraprocessed foods (UPF) has not yet been analyzed in relation to EO-CRC risk, but UPF are associated with westernized diets, which have been identified as a risk factor for the disease (Table 1). Moreover, there is evidence that consumption of UPF is associated with a number of disease risks (Pagliai et al, 2021; Moradi et al, 2023) including cancer (Isaksen and Dankel, 2023), making them an interesting target for future studies of EO-CRC risk and associated microbial and metabolomic alterations (Valicente et al, 2023).

Another significant challenge lies in the integration of genetic, dietary, and microbiome data. The interplay between diet, genetics, and the gut microbiome is intricate and remains incompletely understood. To overcome this, future research should prioritize the incorporation of advanced genomic and microbiome sequencing technologies, allowing for a deeper exploration of how individual genetic and microbial profiles modulate responses to dietary interventions.

A more integrative approach, as proposed by the growing field of molecular pathological epidemiology (MPE), can help identify preventive and therapeutic strategies with improved clinical outcomes for different tumor subtypes or in the context of different host genetic factors. MPE research focusing on *gene-by-environment interactions* has identified genetic biomarkers that predict aspirin's ability to prevent colorectal cancer (Nan et al, 2015). Likewise, an immune-MPE study have shown that omega-3 PUFAs reduce the risk of colorectal carcinoma, specifically for subtypes with high FOXP3+ regulatory T cell counts (Song et al, 2016a). Similarly, a study using a molecular pathological epidemiology database demonstrated a reduced risk of developing CRC with high plasma vitamin D levels depending on the tumor immunity status (Song et al, 2016b). By integrating data from epidemiological studies—including diet, medication, lifestyle, and environmental factors—with classical molecular pathology and gut microbiome data, MPE could help explore the beneficial roles of specific diet–microbiome interactions in tumor subclasses. This will pave the way for tailored dietary recommendations, enhancing preventive strategies against EO-CRC.

In conclusion, while ongoing prospective epidemiological studies and clinical trials are crucial for advancing our understanding of the diet–microbiome relationship in EO-CRC prevention, addressing these challenges will require sustained investment, interdisciplinary collaboration, and innovative research methodologies. By overcoming these obstacles and enhancing the specificity of our investigations, we can develop more effective and personalized dietary interventions, ultimately reducing the global burden of EO-CRC.

## Pending issues

- Explore the protective role of fungi and phages against EO-CRC.
- Develop multi-kingdom biomarkers for risk assessment of EO-CRC.
- Characterization of the diet–microbiome–EO-CRC axis at the micronutrient level.
- Identify dietary habits that prevent the establishment of an oncogenic gut microbiome signature.
- Large-scale epidemiological and clinical trials exclusively targeting EO-CRC.

## Peer review information

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

## Acknowledgements

This work was supported by the German Ministry of Education and Research (BMBF, Project "PerMICCion" 01KD2101) and the Deutsche Forschungsgemeinschaft (DFG, German Research Foundation) under Germany's Excellence Strategy (EXC 2051) project ID 390713860. VD was funded by the German Ministry of Education and Research project 01KD2101D and the Rahel-Goitein-Straus-Program, Medical Faculty Heidelberg University. BG was funded by the ERA-NET on Translational Cancer Research (TRANSCAN), the German Ministry of Education and Research projects 01KT1503 and 01KD2101D, the National Institutes of Health/National Cancer Institute (NHI/NCI) projects R01 CA189184 and U01 CA206110, the Stiftung LebensBlicke, and the Matthias-Lackas Foundations.

## Author contributions

**Zhengyuan Zhou**: Investigation; Visualization; Methodology; Writing—original draft; Writing—review and editing. **Linda Kleis**: Investigation; Visualization; Methodology; Writing—original draft; Writing—review and editing. **Ana Depetris-Chauvin**: Visualization; Methodology; Writing—original draft; Writing—review and editing. **Stefanie Jaskulski**: Investigation; Methodology; Writing—original draft. **Victoria Damerell**: Methodology; Writing—original draft. **Karin B Michels**: Supervision; Writing—review and editing. **Biljana Gigic**: Supervision; Writing—review and editing. **Ute Nöthlings**: Conceptualization; Supervision; Writing—review and editing. **Gianni Panagiotou**: Conceptualization; Supervision; Funding acquisition; Project administration; Writing—review and editing.

## Disclosure and competing interests statement

The authors declare no competing interests.

