## [Peer Review File · EMBO Molecular Medicine]

Beneficial Microbiome and Diet Interplay in Early-Onset Colorectal Cancer

Gianni Panagiotou, Ute Nöthlings, Zhengyuan Zhou, Linda Kleis, Ana Depetris-Chauvin, Stefanie Jaskulski, Karin Michels, Victoria Damerell, and Biljana Gigic

Corresponding authors: Gianni Panagiotou (Gianni.Panagiotou@leibniz-hki.de) , Ute Nöthlings (noethlings@uni-bonn.de)

Review Timeline:

Submission Date:	11th Sep 24
Editorial Decision:	30th Sep 24
Revision Received:	30th Oct 24
Accepted:	8th Nov 24

Editor: Zeljko Durdevic

Transaction Report:

30th Sep 2024

Dear Prof. Panagiotou,

Thank you for the submission of your manuscript to EMBO Molecular Medicine. I am pleased to inform you that we will be able to accept your manuscript pending the following final amendments:

- 1) Please implement referees' suggestions.
- 2) Please upload figures as individual high-resolution files. For Figure 1, please provide detailed description in the figure legend. For figure 2, please increase the size of the icons representing dietary factors and the font of bacterial species. Please place the legends after references.
- 3) Please move tables 1-3 to the main manuscript file and place them after figure legends. Table S1 should be renamed to Appendix Table S1 and uploaded as a PDF file named Appendix with the table of content with page numbers on the title page. Please also update its callout in the main text.
- 4) If BioRender was used to create the figures, please add following sentence to the figure legends: "Graphics were created with BioRender.com."
- 5) Rename "Competing interests declaration" to "Disclosure and competing interests statement". We updated our journal's competing interests policy in January 2022 and request authors to consider both actual and perceived competing interests. Please review the policy <https://www.embopress.org/competing-interests> and update your competing interests if necessary.
- 6) Author contributions: Please remove it from the manuscript and specify author contributions in our submission system. CRediT has replaced the traditional author contributions section because it offers a systematic machine-readable author contributions format that allows for more effective research assessment. Please use the free text boxes beneath each contributing author's name to add specific details on the author's contribution. More information is available in our guide to authors: <https://www.embopress.org/page/journal/17574684/authorguide#authorshipguidelines>
- 7) Glossary: The glossary is meant to explain some of the terms used for laymen. Could you please identify terms that may need an "explanation"?
- 8) Funding: Please make sure that information about all sources of funding are complete in both our submission system and in the manuscript in "Acknowledgments".
- 9) As part of the EMBO Publications transparent editorial process initiative EMBO Molecular Medicine will publish online a Review Process File (RPF) to accompany accepted manuscripts. This file will be published in conjunction with your paper and will include the anonymous referee reports, your point-by-point response and all pertinent correspondence relating to the manuscript. Let us know whether you agree with the publication of the RPF.

I look forward to receiving the revised version of your manuscript.

Yours sincerely,

Zeljko Durdevic

*** IMPORTANT INFORMATION ***

- 1) a .doc formatted version of the manuscript text (including Figure legends and tables)
- 2) Separate figure files
- 3) a letter INCLUDING the reviewer's reports and your detailed responses to their comments.

Also, and to save some time should your paper be accepted, please read below for additional information regarding some features of our research articles:

1) Glossary: EMBO Molecular Medicine articles will be accompanied by a glossary explaining some of the terms used for laymen. I identified the following:

_____, _____, _____

Could you please help us in identifying terms that may need an "explanation" other terms that we can add to the glossary.

2) For more information: This is a short list of related web links for further consultation by the readers. Could you identify some relevant ones? Examples are patient associations, OMIM related links, databases, authors websites, etc.

3) Pending issues: At the end of each article we will have a box highlighting issues that still need further studies and where research efforts should converge (we call this the Pending issues box). From my reading I would say:

but I can see there may be many more. Could you work on this as well?

4) Disclosure and competing interest statement: Please include a statement declaring any competing commercial interests in relation to your submitted work.

5) Please note that we now mandate that all corresponding authors list an ORCID digital identifier. This takes <90 seconds to complete. We encourage all authors to supply an ORCID identifier, which will be linked to their name for unambiguous name identification.

Currently, our records indicate that the ORCID for your account is 0000-0001-9393-124X.

Link Not Available

-

Thank you,

Zeljko Durdevic

***** Reviewer's comments *****

Referee #1 (Remarks for Author):

The manuscript addresses an important and timely topic, exploring the complex relationship between diet, beneficial gut microbiome, and early-onset colorectal cancer (EO-CRC). The increasing incidence of EO-CRC globally, especially among younger populations, highlights the need for an in-depth understanding of its potential risk factors and preventive strategies. The authors have done an excellent job providing a comprehensive overview of the dietary factors involved in EO-CRC development and the protective role of specific gut microbiota. The structure is logical, and the manuscript flows well from one section to another.

While the paper thoroughly covers the protective role of various microbial species, the mechanisms by which these species influence EO-CRC could be explained in greater detail. The manuscript mentions apoptosis induction, immune modulation, and proliferation inhibition but does not always connect these mechanisms clearly to specific microbial taxa.

Besides , although the manuscript mentions EO-CRC frequently, many of the cited studies and findings seem to pertain to

colorectal cancer (CRC) more generally. While it is reasonable to draw parallels between EO-CRC and CRC in older adults, it would be beneficial to clearly differentiate between findings specific to EO-CRC and those that pertain to CRC overall. Finally, the manuscript provides valuable insight into diet-related risk factors, but maybe the explanation of how specific dietary components influence the microbiome (and subsequently affect EO-CRC risk) could be further developed.

Referee #2 (Remarks for Author):

Contents are generally good and well written. This paper discusses diet and microbiome in EO-CRC, which are very important.

This paper should discuss following links: link #1 between Western dietary pattern and the incidence of CRC containing abundant pks+ *Escherichia coli* (Arima et al. *Gastroenterology* 2022); link #2 between pks+ *E. coli* and SBS88 colibactin-induced mutational signature in CRC (Clever lab, *Nature* 2020); link #3 between colibactin-induced mutational signature and EO-CRC (Rosendahl Huber et al. *Cancer Cell* 2024; Georgeson et al. *MedRxiv*). These studies collectively nicely make likely causal links from western diet (risk factor) -> pks+ *E. coli* (microbe) -> SBS88 mutational signature (pathogenic mechanism) -> EO-CRC development, thereby providing enormous etiologic insight.

Related to the above point, research on lifestyle, environment, and other factors should be integrated with analyses of personalized tumor biomarkers. The authors should discuss molecular pathological epidemiology research that can investigate those variables. Molecular pathological epidemiology research has been discussed in the literature, eg, *Ann Rev Pathol* 2019, etc and can be a promising direction. This will also significantly add novelty to this article.

Dear Dr Zeljko Durdevic,

We are very pleased to submit a revised version of our manuscript entitled "Beneficial Microbiome and Diet Interplay in Early-Onset Colorectal Cancer." We have carefully analyzed all the referee's comments and are confident that we have thoroughly addressed all their concerns.

Below, we present each referee's comment, followed by our response in blue.

Response to referee's comments

Referee #1 (Remarks for Author):

The manuscript addresses an important and timely topic, exploring the complex relationship between diet, beneficial gut microbiome, and early-onset colorectal cancer (EO-CRC). The increasing incidence of EO-CRC globally, especially among younger populations, highlights the need for an in-depth understanding of its potential risk factors and preventive strategies. The authors have done an excellent job providing a comprehensive overview of the dietary factors involved in EO-CRC development and the protective role of specific gut microbiota. The structure is logical, and the manuscript flows well from one section to another.

While the paper thoroughly covers the protective role of various microbial species, the mechanisms by which these species influence EO-CRC could be explained in greater detail. The manuscript mentions apoptosis induction, immune modulation, and proliferation inhibition but does not always connect these mechanisms clearly to specific microbial taxa.

Besides although the manuscript mentions EO-CRC frequently, many of the cited studies and findings seem to pertain to colorectal cancer (CRC) more generally. While it is reasonable to draw parallels between EO-CRC and CRC in older adults, it would be beneficial to clearly differentiate between findings specific to EO-CRC and those that pertain to CRC overall.

We sincerely thank the referee for their positive valuation of our manuscript. Details on the different mechanisms by which microbial species influence CRC are presented in the subsection "Molecular mechanisms implicated in gut microbiome beneficial effects" and Table II. We have added more mechanistic details to the section, which is presented with Track Changes (•page 7, lines 284-301, •page 8, lines 306-317, lines 333-340). For clarity, we have added a column to Table II indicating whether the mechanism is classified as "Apoptosis and proliferation inhibition," "Immune system regulation," or "Microbial interactions and synergistic therapeutic effects."

We agree with the reviewer that many of the cited studies in the subsection "Molecular mechanisms implicated in gut microbiome beneficial effects" which reflect a limitation of most mechanistic studies not considering the age of disease of onset in their models. We have now explicitly mentioned this caveat and the implications for understanding the underlying mechanisms specifically implicated in EO-CRC in our manuscript (•page 9, lines 387-391).

Finally, the manuscript provides valuable insight into diet-related risk factors, but maybe the explanation of how specific dietary components influence the microbiome (and subsequently affect EO-CRC risk) could be further developed.

Following the referee's suggestion, we have now provided some explanations of how specific dietary components could influence the gut microbiome in the section "Specific associations of diet and beneficial microbes in CRC" (•page 11, lines 438-443, lines 453-456, 467-470, and 473-476, •page 12, lines 485-489, lines 492-500, and lines 517-519).

Referee #2 (Remarks for Author):

Contents are generally good and well written. This paper discusses diet and microbiome in EO-CRC, which are very important.

We deeply appreciate the referee's interest in our manuscript.

This paper should discuss following links:

link #1 between Western dietary pattern and the incidence of CRC containing abundant pks+ Escherichia coli (Arima et al. Gastroenterology 2022);

link #2 between pks+ E. coli and SBS88 colibactin-induced mutational signature in CRC (Clever lab, Nature 2020);

link #3 between colibactin-induced mutational signature and EO-CRC (Rosendahl Huber et al. Cancer Cell 2024; Georgeson et al. MedRxiv).

These studies collectively nicely make likely causal links from western diet (risk factor) -> pks+ E. coli (microbe) -> SBS88 mutational signature (pathogenic mechanism) -> EO-CRC development, thereby providing enormous etiologic insight.

We thank the referee for pointing to these studies, which indeed represent a convincing body of work on the causal link between Western Diet, gut microbes, mechanism, and EO-CRC. We have now included these publications when we present the association between a Western diet and EO-CRC in the section "Specific associations of diet and beneficial microbes in CRC" (page 12, lines 510-514).

Related to the above point, research on lifestyle, environment, and other factors should be integrated with analyses of personalized tumor biomarkers. The authors should discuss molecular pathological epidemiology research that can investigate those variables. Molecular pathological epidemiology research has been discussed in the literature, eg, Ann Rev Pathol 2019, etc and can be a promising direction. This will also significantly add novelty to this article.

We agree with the referee that molecular pathological epidemiology (MPE) research is of great interest for the topic of EO-CRC in general and particularly in its relationship with diet and the gut microbiome. We have now included a paragraph on the opportunities that such an integrative approach could offer to CRC management in the section "Current challenges and future directions" (page 15, lines 613-624). We focused only on the possibilities of improving prevention strategies and not on diagnosing or developing personalized tumor biomarkers (as suggested by the referee) because those aspects are beyond the scope of our review article.

8th Nov 2024

Dear Prof. Panagiotou,

We are pleased to inform you that your manuscript is accepted for publication and is now being sent to our publisher to be included in the next available issue of EMBO Molecular Medicine.

Your manuscript will be processed for publication by EMBO Press. It will be copy edited and you will receive page proofs prior to publication.

You will soon be contacted by our publisher Springer Nature to sign your publishing license. When you login to the customer service website, please use the token/code copied below to waive the article publication charges. Should you experience any difficulty, please email publishing@embo.org.

Waiver token: XXXXXXXXXXXXXXXXXXXX
